# Cordierite-Supported Transition-Metal-Oxide-Based Catalysts for Ozone Decomposition

**Maria Chernykh, Maria Grabchenko *, Alexey Knyazev and Grigory Mamontov ***

Chemical Faculty, Tomsk State University, 634050 Tomsk, Russia; msadlivskaya@mail.ru (M.C.); kas854@mail.ru (A.K.)
* Correspondence: marygra@mail.ru (M.G.); grigoriymamontov@mail.ru (G.M.)

**Abstract:** Cordierite-based supported noble-metal-free catalysts for ozone decomposition are elaborated. The cordierite ceramic surface is pretreated with oxalic acid and NaOH, and Mn-Cu-Ni oxide catalysts are prepared by the impregnation method. The mass ratio of the supported oxides in the resulting catalysts is $MnO_2$:$CuO$:$NiO$ = 3:2:1, and their loadings are from 1.8 to 7.0 wt.%. The pretreated supports and catalysts are characterized by low-temperature $N_2$ adsorption, scanning electron microscopy (SEM), powder X-ray diffraction analysis (XRD), and temperature-programmed reduction with $H_2$ (TPR-$H_2$). The catalysts are tested in ozone decomposition with high airflow rates (20 and 50 L/min) and with initial ozone concentrations of 1 and 2 ppm at temperatures in the range of 25–120 °C. It is shown that a combined treatment of cordierite with oxalic acid and NaOH leads to a developed porous structure and stabilization of supported Mn-Cu-Ni oxides in a highly dispersed state. The high activity of catalysts in ozone decomposition at room temperature and high airflow is demonstrated. The developed catalysts can be recommended for application in purification of air from the ozone because of their high catalytic activity, high mechanical stability, and relatively low weight and cost.

**Keywords:** ozone decomposition; cordierite; manganese oxides; honeycomb catalysts; hierarchical porous structure; noble-metal-free catalyst





## 1. Introduction

Ozone is one of the most harmful air pollutants [1]. Long-term ozone exposure can lead to serious impairment of human health (dizziness, inflammation of the mucous membranes and respiratory tract, and even premature mortality) and the environment (reduction in crop yields, contribution to global warming, etc.) [2]. Ozone sources are divided into outdoor and indoor [3]. The predominant outdoor ozone sources are photochemical reactions in air between volatile organic compounds (VOCs) and nitrogen oxides ($NO_x$) [4]. Most indoor ozone comes from outdoor sources; however, there is a number of equipment producing ozone during their operation and, thus, polluting the air (air purifiers, copiers, air conditioners, sterilizers, etc.) [5]. Rapid indoor air purification from ozone is an urgent challenge since people are indoors most of the time.

Catalytic decomposition is the most energetically and economically efficient method of ozone decomposition (activated carbon adsorption, thermal decomposition, chemical absorption, etc.) [6]. The nature of ozone decomposition catalysts is diverse. For instance, an aircraft cabin ozone decomposer contains the $Pd/Al_2O_3$ catalyst, while $MnO_2/AC$ is used in photocopiers [7]. Noble-metal-based catalysts feature excellent activity in catalytic ozone decomposition, but they have limited practical application due to their high cost. Therefore, currently, catalysts based on transition metal oxides or even their mixtures are widely used [8]. Such catalysts feature a lower cost compared with supported catalysts based on noble metals, but they are not inferior to them in catalytic activity due to their redox activity, variable valence states, and good stability.

It was shown by S. Ted Oyama et al. [7] that $MnO_2$ demonstrates the highest activity in ozone decomposition among the transition metal oxides. However, since the manganese atom has the $3d^5 4s^2$ electronic configuration of the outer energy level, it can exist in various oxide states such as $MnO_2$, $MnO$, $Mn_2O_3$, $Mn_3O_4$, and $Mn_5O_8$; therefore, there are many works devoted to the study of $MnO_x$ catalytic activity in ozone decomposition [9–12]. The high activity of the manganese oxides in the $O_3$ decomposition is due to the redox ability of $Mn^{n+}$ to be converted into $Mn^{n+1}$ [6]:

$$O_3 + \left[Mn^{n+}\right] \rightarrow O_2 + O^- \left[Mn^{n+1}\right] \tag{1}$$

$$O_3 + O^- \left[Mn^{n+1}\right] \rightarrow 2O_2 + \left[Mn^{n+}\right] \tag{2}$$

The attention of scientists has recently been increasingly attracted by the catalytic studies of mixed manganese oxides with other transition metals (Ni, Cu, etc.) since the addition of such oxides increases the redox ability of the manganese oxide. For instance, J. Ma et al. [13] compared the 40 ppm ozone conversion on $\alpha$-$Mn_2O_3$ and $CeMn_{10}O_x$ catalysts and showed that the addition of Ce increased the ozone decomposition activity by 2.5 times. D. Mehandjiev et al. [14] showed high activity of $NiMnO_3$-ilmenite and $NiMn_2O_4$-spinel catalysts in ozone decomposition, benzene, and CO oxidation. J. Kim et al. [15] also showed that Cu, Fe, and Ru additives to the Mn/HZSM-5 catalyst led to an increase in its catalytic efficiency. Thus, the study of the catalytic activity of materials based on oxides of manganese and other transition metals to identify the most active compositions is an urgent challenge.

Cordierite ($2MgO\cdot 2Al_2O_3\cdot 5SiO_2$) is a promising support since it is a rather convenient system for a wide range of practical applications, usually in the honeycomb configuration [16–19]. Cordierite features high thermal stability (decomposition temperature, 1640 °C) and a low thermal expansion coefficient ($1.8 \cdot 10^{-6}$ °C$^{-1}$), which allows it to be used as a stable base in catalysts for various high-temperature processes [20]. Since the specific surface area of the cordierite is rather small, a secondary support, e.g., $Al_2O_3$ or $SiO_2$, is usually introduced onto the cordierite surface from the suspension [21,22]. Despite the small increase in the specific surface area, the mass of the sample increases and the applied layer does not adhere well; in particular, the resistance to vibration of such samples is rather low. All this makes such materials unattractive for real operating conditions. Another approach is cordierite etching, which results in an increase in the defectiveness of the cordierite structure due to the leaching of metal ions from cordierite with an increase in the specific surface area and porosity. This pretreatment reduces the weight of the final catalyst and facilitates strong binding of the supported active components to the cordierite surface.

Therefore, the aim of this work was to choose cordierite support pretreatment conditions, synthesize and characterize the transition metal oxide (Mn, Cu, and Ni) catalysts on the basis thereof, and study their catalytic properties in ozone decomposition.

## 2. Materials and Methods

### 2.1. Synthesis of Supports

All reagents of high chemical purity were used. Cordierite blocks (Jiangsu Yixing Nonmetallic Chemical Machinery Factory Co., Ltd., Dingshu Town, China) with a diameter of 150 mm, height of 150 mm, and cell density of 400 cells per square inch were used. The cylindrical blocks with a diameter of 22 mm and length of 50 mm were fine-cut and used for the catalyst preparation (Figure 1). The blocks were washed three times with deionized water; purged with air; and treated in aqueous solutions of $HNO_3$, $H_2C_2O_4$, or NaOH at 70–90 °C for 24 h to increase the surface area (see conditions of pretreatment in Table 1). The samples were labeled Cor-$HNO_3$, Cor-0.2$H_2C_2O_4$, Cor-1$H_2C_2O_4$, Cor-2$H_2C_2O_4$, Cor-4$H_2C_2O_4$, and Cor-NaOH, respectively. A cordierite block pretreated in $H_2C_2O_4$ and NaOH solutions was labeled as Cor-optimized. Then, all samples were washed in deionized water three times, purged with air, and dried at 120 °C for 12 h.

The value of weight loss (Table 1) is associated with the leaching of components from the cordierite block by acid or/and alkali.

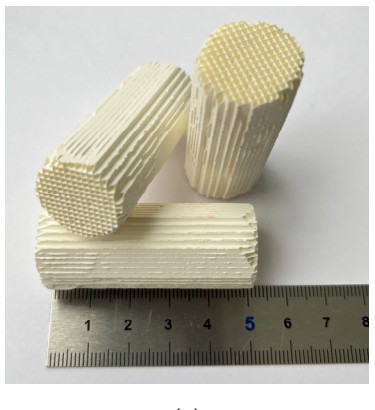

(**a**)

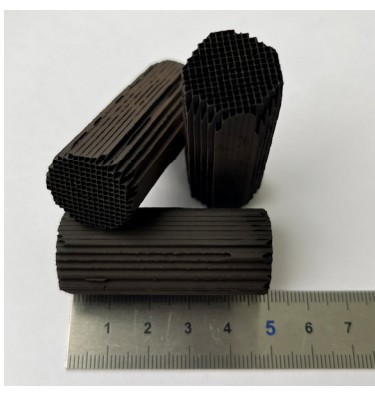

(**b**)

**Figure 1.** Initial fine-cut cordierite blocks (**a**) and synthesized catalysts (**b**).

**Table 1.** Conditions of support pretreatment.

| Sample | Pretreatment | Weight Loss (%) |
|---|---|---|
| Cor-HNO$_3$ | 1.85 mol/L HNO$_3$, 70 °C, 24 h | 9.1 |
| Cor-NaOH | 2 mol/L NaOH, 90 °C, 24 h | 3.6 |
| Cor-0.2H$_2$C$_2$O$_4$ | 0.2 mol/L H$_2$C$_2$O$_4$, 70 °C, 24 h | 4.8 |
| Cor-1H$_2$C$_2$O$_4$ | 1 mol/L H$_2$C$_2$O$_4$, 70 °C, 24 h | 10.5 |
| Cor-2H$_2$C$_2$O$_4$ | 2 mol/L H$_2$C$_2$O$_4$, 70 °C, 24 h | 14.4 |
| Cor-4H$_2$C$_2$O$_4$ | 4 mol/L H$_2$C$_2$O$_4$, 70 °C, 24 h | 17.7 |
| Cor-optimized | Combination of NaOH and H$_2$C$_2$O$_4$ | 23.9 |

### 2.2. Synthesis of Catalysts

To prepare the catalysts, Cor-2H$_2$C$_2$O$_4$ and Cor-optimized were chosen as supports with the highest characteristics. The support blocks were impregnated with an aqueous solution of precursors, i.e., Mn(NO$_3$)$_2$·6H$_2$O, Cu(NO$_3$)$_2$·3H$_2$O, and Ni(NO$_3$)$_2$·6H$_2$O (chemically pure, REACHEM, Moscow, Russia). Impregnation was carried out in excess of the impregnating solution with a single and double immersion in the solution (samples Cat-imp1 and Cat-imp2, respectively). Thermal treatment of the samples after impregnation included drying at 120 °C for 12 h and calcination at 500 °C for 4 h. The values of weight gain show that the content of supported oxides was 2.0 wt.% and 3.8 wt.% for Cat-imp1 and Cat-imp2, respectively.

The impregnation under vacuum was applied for better infiltration of the impregnating solution into the cordierite pores. The support was immersed in the Bunsen flask and then evacuated to a pressure of ~700 mbar. The excess of the impregnating solution was added under vacuum; then, the pressure was increased up to the atmospheric one. Next, the sample was dried and calcined. The loading of active components for catalysts prepared by vacuum impregnation (sample Cat-vac.imp) was 1.8 wt.%. The Cat-optimized sample was prepared by impregnation of the Cor-optimized support (the loading of active components was 7.0 wt.%). The mass ratio of the supported oxides in the resulting catalysts was MnO$_2$:CuO:NiO = 3:2:1.

### 2.3. Materials Characterization

The specific surface area and porosity were studied by low-temperature nitrogen adsorption–desorption at −196 °C on the 3Flex device (Micromeritics, Norcross, GA, USA). All samples were degassed under vacuum (10$^{-2}$ Torr) at 200 °C for 2 h at the VacPrep Degasser degassing station (Micromeritics, USA). The specific surface area was determined using the multipoint BET method by straightening the adsorption isotherm in the p/p$_o$

range from 0.05 to 0.30. The BJH-Adsorption method was used to calculate the pore size distributions, and the t-plot method was used to estimate the surface area and volume of micropores.

The mechanical properties of the obtained samples were measured on the "Measuring device to determine the granule strength IPG 1M" (LLC "UNIKHIM", Ekaterinburg, Russia). A small block with a cubic shape and a side length of 5–8 mm was broken, and the strength was calculated as the force-to-square ratio: P(MPa) = F(N)/(length $*$ wight) mm$^2$.

The sample resistance to vibration as an adhesion of the active components to the support was studied using an ultrasonic bath ("Sapphire", Russia) according to the procedure described in Ref. [23]. The samples were placed in acetone and exposed to ultrasound (35 kHz and 50 W) for 30 min at room temperature. Five consecutive cycles were carried out. The resistance was calculated as the weight of a block after such a treatment.

The phase composition of the samples was determined by XRD on the XRD-7000 (Shimadzu, Kyoto, Japan) using the CuK$\alpha$ radiation (1.54 Å) in the range of angles 2θ of 10–80° and a scanning rate of 0.02 °/s. The phase composition was analyzed using the PDF 4+ databases as well as the POWDER CELL 2.4 full-profile analysis program. The morphology of the support and catalysts was studied by SEM on the Quanta 200 3D (EDAX, Netherlands) at an acceleration voltage of 30 kV.

The method of temperature-programmed reduction with hydrogen (TPR-H$_2$) was used to determine the features of the reduction of the supported oxides (MnO$_2$, CuO, NiO) using the AutoChem HP analyzer (Micromeritics, Norcross, USA) in a 10% H$_2$/Ar gas mixture flow during the linear heating from 25 to 900 °C with a heating rate of 10 °C/min.

### 2.4. Ozone Decomposition

The catalytic characteristics of the synthesized samples were studied in the ozone decomposition reaction on the catalytic unit "Ozon" (LLC "Kvarta", Novosibirsk, Russia). A UV-lamp DBK-18 (Polus, Moscow, Russia) with an ozone-generating line at 185.6 nm in its emission spectrum was used as an ozone source. The initial ozone concentration was 1-2 ppm, which is similar to the possible maximal ozone concentration in the atmosphere at heights of up to 10 km at which aircrafts fly [24]. The catalytic layer length was 150 mm (3 blocks with a length of 50 mm, Figure 1). The reaction was carried out at high airflow rates of 20 L/min and 50 L/min that correspond to a GHSV (gas hourly space velocity) of ~25,500 and 63,700 h$^{-1}$, respectively. The most active catalyst was tested under harder conditions at an airflow rate of 150 L/min with heating from 25 to 120 °C. Such conditions of catalyst testing are similar to the ones in air purification systems. The change in the ozone concentration at the reactor inlet and outlet was monitored by the "Sigma-O3.DE" sensors (LLC "Prompribor-R", Moscow, Russia).

## 3. Results and Discussion

### 3.1. Optimization of the Support Pretreatment Conditions

Since the initial cordierite ceramic is not a porous material, there are various approaches to increase the porosity and specific surface area of the cordierite blocks. The deposition of a secondary support (usually silica or alumina) from sol on the cordierite surface is usually used to increase the surface area of the ceramic support [22,25]. This leads to an increased catalyst weight and low coating stability. In the case of catalyst application for ozone decomposition in aircraft, both the catalyst weight and high mechanical stability towards vibration are crucial. Thus, in the present work, we applied the chemical cordierite treatment by acids and alkali to increase the surface area and decrease the catalyst weight. Figure 2 and Table 2 show the results of low-temperature N$_2$ adsorption–desorption for the initial cordierite block and the one after treatment by acids and alkali. Almost no nitrogen adsorption is observed for untreated cordierite (Figure 2a, sample "initial cordierite"); therefore, it has a specific surface area below 1 m$^2$/g (Table 2).

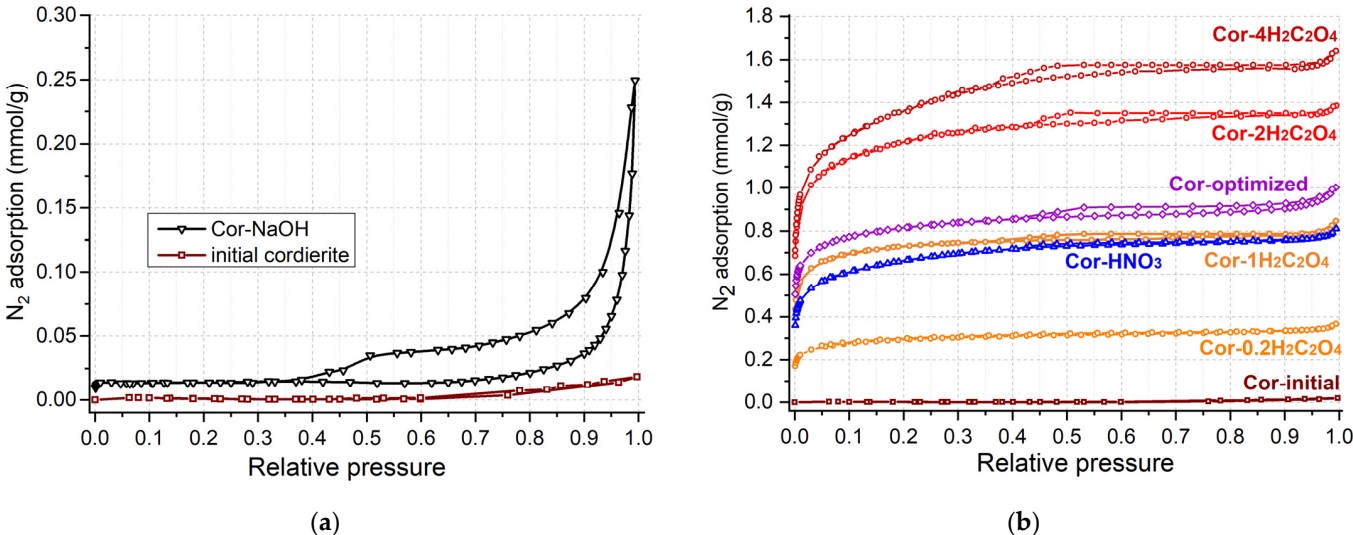

**Figure 2.** Isotherms of $N_2$ adsorption–desorption for cordierite etched with alkali (**a**), and with $HNO_3$ and $H_2C_2O_4$ with different concentrations (**b**).

**Table 2.** Characteristics of cordierite supports.

| Sample | $S_{BET}$ (m²/g) | $S_{micro}$ [1] (m²/g) | $V_{pore}$ (cm³/g) | $V_{micro}$ [1] (cm³/g) | Strength (MPa) |
|---|---|---|---|---|---|
| Initial cordierite | 0.04 | - | - | - | 2.7 |
| Cor-HNO₃ | 54 | 32 | 0.028 | 0.013 | 3.1 |
| Cor-NaOH | 1.5 | - | 0.008 | 0.001 | 2.8 |
| Cor-0.2H₂C₂O₄ | 24 | 17 | 0.012 | 0.007 | 1.8 |
| Cor-1H₂C₂O₄ | 56 | 37 | 0.029 | 0.015 | 2.3 |
| Cor-2H₂C₂O₄ | 99 | 66 | 0.047 | 0.028 | 1.9 |
| Cor-4H₂C₂O₄ | 107 | 56 | 0.055 | 0.025 | 1.0 |
| Cor-optimized | 68 | 50 | 0.034 | 0.020 | 0.4 |

[1] According to the t-plot-method.

A hysteresis loop in the relative pressure range of 0.5–1 on the isotherm of $N_2$ adsorption–desorption is observed for cordierite treated with 2M NaOH solution (sample Cor-NaOH), which indicates the formation of meso- and macropores. The dissolution of the silica component from the cordierite should be predominant under alkali conditions:

$$SiO_2 + 2\,NaOH \leftrightarrow Na_2SiO_3(aq) + H_2O \qquad (3)$$

Leaching of the alumina component is also possible:

$$Al_2O_3 + 2\,NaOH + 3\,H_2O \leftrightarrow 2\,Na[Al(OH)_4](aq) \qquad (4)$$

The specific surface area and pore volume of the Cor-NaOH sample increased insignificantly, reaching values of 1.3 m²/g and 0.006 cm³/g, respectively (Table 2). Since the weight loss after the treatment in alkali amounted to only 3.6 wt.% (Table 1), the mechanical strength of the sample is commensurate with that of the initial support (2.8 MPa, Table 2). Thus, it can be concluded that cordierite treatment in the alkali solution leads to the formation of wide meso- and macropores without a significant rise in $S_{BET}$.

The cordierite treatment with acids (oxalic or nitric) led to a growth in the adsorption value at a low relative pressure (Figure 2b), which indicates the formation of micro- and small mesopores in the cordierite structure. The Cor-HNO₃ and Cor-1H₂C₂O₄ samples have similar specific surface area values of 54 and 56 m²/g, and a total pore volume of 0.028 and 0.029 cm³/g (Table 2), respectively, which indicates a similarly efficient support etching process.

The dissolution of alumina and magnesia under acid conditions occurs, while silica is stable in both solutions of nitric and oxalic acids:

$$Al_2O_3 + 6\ HNO_3 \rightarrow 2\ Al(NO_3)_3(aq) + 3\ H_2O \tag{5}$$

$$MgO + 2\ HNO_3 \rightarrow Mg(NO_3)_2(aq) + H_2O \tag{6}$$

$$Al_2O_3 + 3\ H_2C_2O_4 \rightarrow Al_2(C_2O_4)_3(aq) + 3\ H_2O \tag{7}$$

$$MgO + H_2C_2O_4 \rightarrow MgC_2O_4(aq) + H_2O \tag{8}$$

Since the products of nitric acid decomposition are more harmful to the environment and people, oxalic acid was chosen for subsequent experiments. Figure 2b shows the isotherms of $N_2$ adsorption–desorption for supports treated with oxalic acid solutions with various concentrations. With an increase in the concentration of the oxalic acid solution from 0.2 mol/L to 4 mol/L, an increase in nitrogen adsorption is observed and, accordingly, an increase in the specific surface area from 24 to 107 $m^2$/g and the pore volume from 0.012 to 0.055 $cm^3$/g, respectively. However, it can be seen that the efficiency of the support etching decreases upon passing from a solution with concentrations of 2 mol/L to 4 mol/L (Table 2). The support fragility increases with an increase in the concentration of oxalic acid since the weight loss rises up to 17.7 wt.% and the mechanical strength decreases to 1.0 MPa for Cor-4$H_2C_2O_4$. Thus, the cordierite etching with oxalic acid leads to a significant increase in the specific surface, but the micropores are predominantly formed. Since the Cor-2$H_2C_2O_4$ achieves a high specific surface area and pore volume while maintaining a significant mechanical strength (1.9 MPa), further synthesis of catalysts was carried out using the Cor-2$H_2C_2O_4$ sample as a support.

Since the formation of both micro- and mesopores in the support is important for better stabilization of the supported active components, the Cor-optimized support was prepared by combined etching with solutions of $H_2C_2O_4$ and NaOH. It can be seen from the SEM images for initial cordierite (Figure 3a,b) that it is characterized only by a macroporous structure.

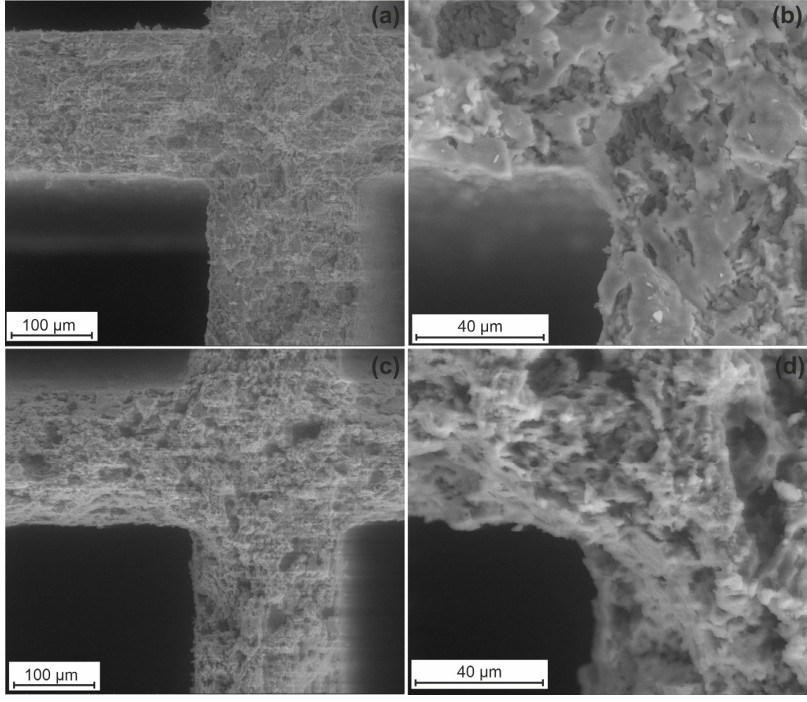

**Figure 3.** SEM images of initial cordierite (**a**,**b**) and Cor-optimized support (**c**,**d**).

The formation of macropores is attributed to the preparation of the cordierite block by the extrusion method. Cordierite etching with solutions of $H_2C_2O_4$ and NaOH leads to a rise in the specific surface area of up to 68 $m^2/g$ and the pore volume of up to 0.034 $cm^3/g$. Figure 3c,d demonstrate the SEM images for the Cor-optimized support. The structure density decreases after the treatment with $H_2C_2O_4$ and NaOH, and more pores are formed.

Thus, the mechanisms of acid and alkali pretreatments are different. The leaching of Al and Mg under acid conditions leads to the formation of micropores, while the Si leaching from cordierite under alkali conditions leads to the formation of meso- and macropores. Probably, the alkali treatment occurs with the transport of soluble silica ($Na_2SiO_3$(aq)) both through the solution and along the surface, which is also accompanied by the re-deposition of silica on the cordierite surface.

### 3.2. Textural and Structural Properties of the Catalysts

Figure 4 shows the $N_2$ adsorption–desorption isotherms and pore size distributions of the synthesized catalysts. Isotherms are characterized by the increase in the adsorption value at low relative pressures, which indicates the predominant formation microporous structure. The presence of a hysteresis loop in the range of relative pressures of 0.45–0.9 indicates the capillary condensation in thin mesopores. The shape of the hysteresis loop (according to the IUPAC, it is H4 [26]) also indicates the presence of micropores and thin mesopores for the samples prepared on the Cor-2$H_2C_2O_4$ support (Figure 4a).

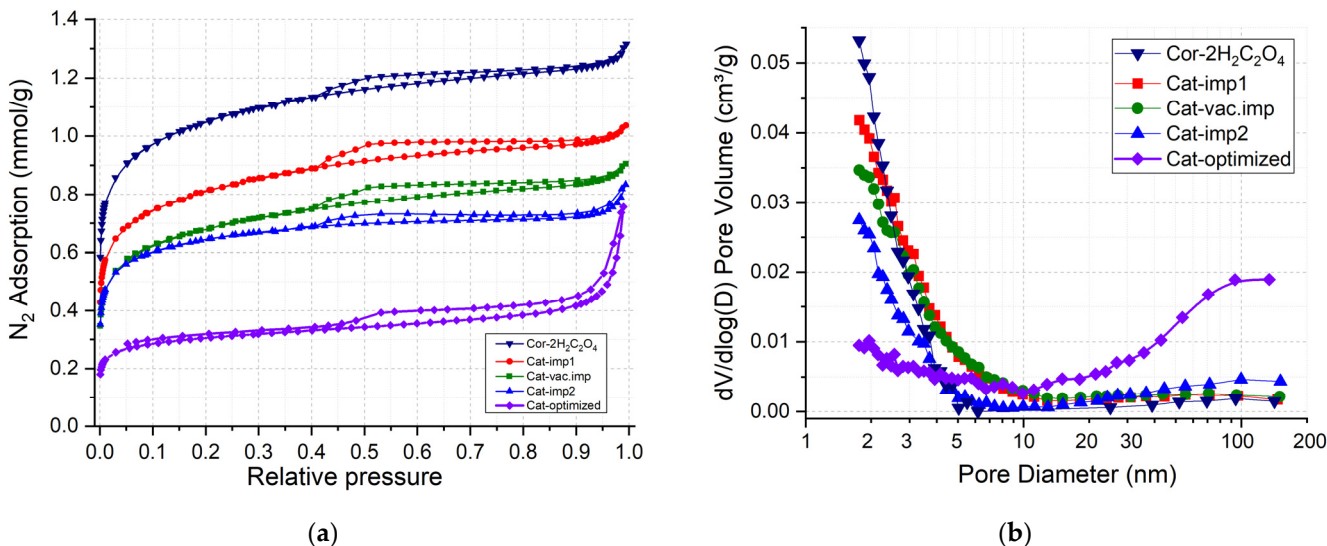

(**a**)　　　　　　　　　　　　　　　　(**b**)

**Figure 4.** Isotherms of $N_2$ adsorption–desorption (**a**) and corresponding pore size distributions (**b**) for Cor-2$H_2C_2O_4$ support and prepared catalysts.

The pore size distribution curves (Figure 4b) show that all catalysts are characterized by pores with sizes up to 8 nm. The deposition of the Mn-Cu-Ni oxides on the surface of the COR-2$H_2C_2O_4$ support leads to a filling of micropores and thin mesopores that is accompanied by a decrease in the $S_{BET}$ and in the pore volume from 99 to 53 $m^2/g$ and from 0.047 to 0.029 $cm^3/g$, respectively (Table 3).

The Cat-optimized sample with a loading of deposited oxides of 7 wt.% was prepared by impregnation of the Cor-optimized support. The specific surface area and pore volume decreased from 68 $m^2/g$ and 0.034 $cm^3/g$ for Cor-optimized to 25 $m^2/g$ and 0.018 $cm^3/g$ for Cat-optimized (Table 3). It is noteworthy that the catalyst prepared under the optimized conditions features micro-, meso-, and macropores (Figure 4a,b). Thus, a combination of cordierite pretreatment with oxalic acid and NaOH and supporting of Mn-Cu-Ni oxides leads to the formation of catalysts with a hierarchical porous structure.

**Table 3.** Characteristics of catalysts.

| Sample | $S_{BET}$ (m²/g) | $S_{micro}$ [1] (m²/g) | $V_{pore}$ (cm³/g) | $V_{micro}$ [1] (cm³/g) | $\omega(MeO_x)$ (%) | Strength to Vibration (%) | Strength (MPa) |
|---|---|---|---|---|---|---|---|
| Cat-imp1 | 63 | 32 | 0.036 | 0.014 | 2.0 | 100 | 2.4 |
| Cat-vac.imp | 53 | 27 | 0.031 | 0.012 | 1.8 | 99.9 | 1.7 |
| Cat-imp2 | 54 | 35 | 0.029 | 0.014 | 3.8 | 98.9 | 1.7 |
| Cat-optimized | 25 | 16 | 0.018 | 0.007 | 7.0 | 99.9 | 1.1 |

[1] According to the t-plot-method.

The mechanical properties of the catalysts were measured as the strength towards breaking under force and vibration. The catalysts prepared based on the Cor-2H$_2$C$_2$O$_4$ support are characterized by similar values of mechanical strength of 1.7–2.4 MPa. The Cat-optimized catalyst had a mechanical strength of 1.1 MPa, while the one for the Cor-optimized support was only 0.4 MPa. Thus, support of the active component for this catalyst leads to a significant growth in the strength. The strength under vibration for the catalysts is 99.9–100%, which indicates high component adhesion on the surface of the cordierite support. The decrease in strength for the Cat-imp2 catalyst to 98.9% indicates the decreasing adhesion of the active oxide precursors from the second portion of the impregnating solution during the double impregnation of the support. Thus, the impregnation technique is favorable for the stabilization of the active oxide on the cordierite surface. For comparison, in Ref. [21], support of the CeO$_2$ nanoarray on the surface of cordierite was carried out by the hydrothermal method, and the strength to vibration was only 96% (the CeO$_2$ loading was 5.2% wt.).

The features of porous structure and active components distribution in the Cat-optimized catalysts were studied by SEM accompanied by the EDX analysis (Figures 5 and 6). Figure 5a,b show that the catalyst is characterized by a texture similar to the one for the Cor-optimized support (Figure 3c,d). A uniform distribution of the active components (Mn, Cu, and Ni) can be seen from the corresponding maps of their distribution (Figure 5c–f).

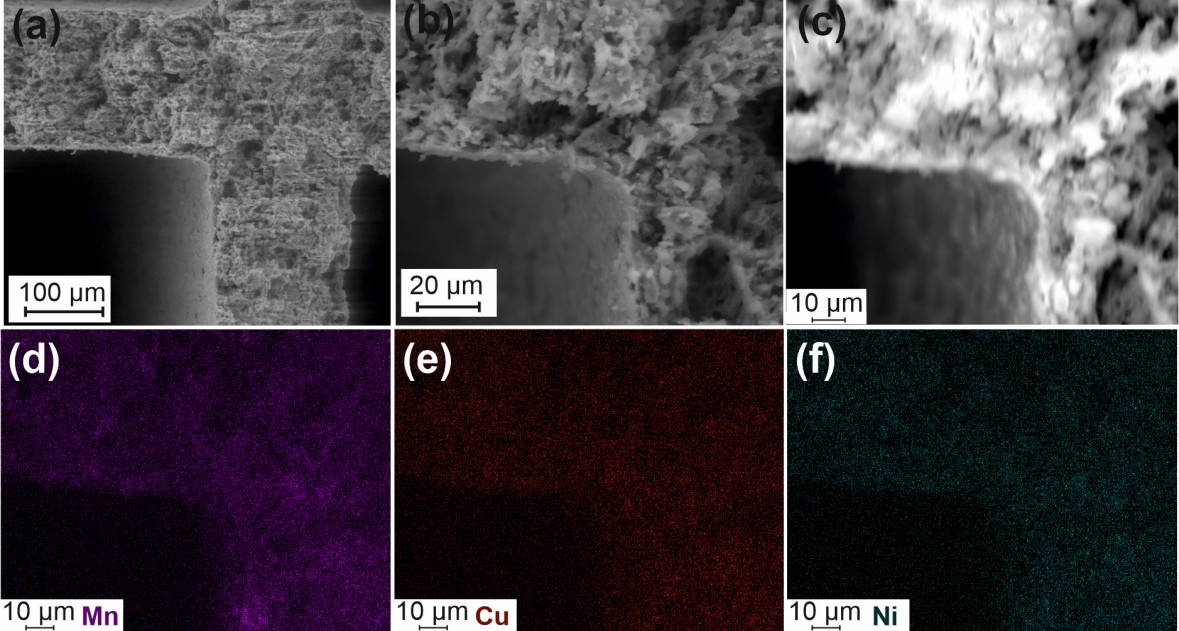

**Figure 5.** SEM images for Cat-optimized (**a**–**c**) and mapping for Mn (**d**), Cu (**e**), and Ni (**f**).

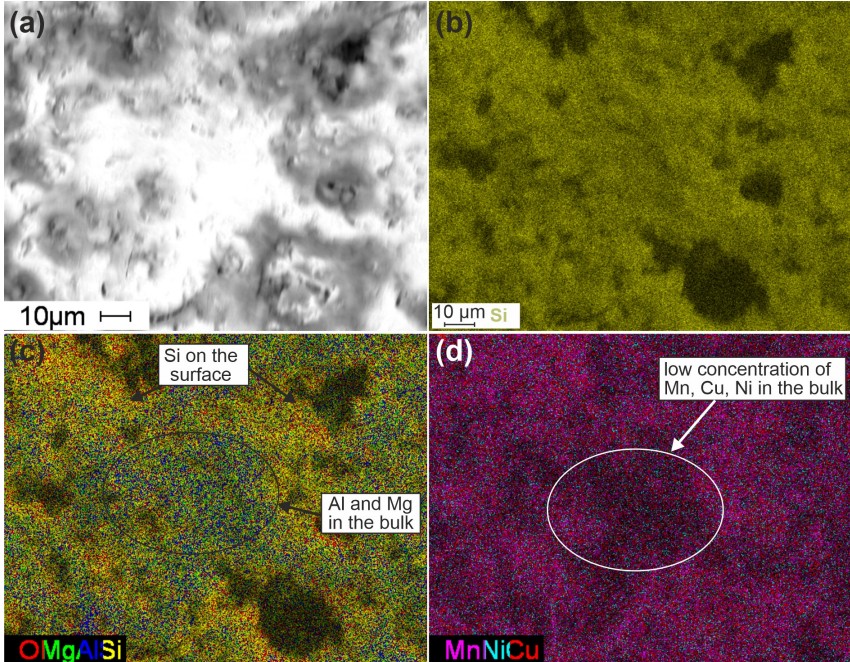

**Figure 6.** SEM images of Cat-optimized catalysts (**a**) and mapping for Si (**b**); O, Mg, Al, and Si (**c**); and Mn, Ni, and Cu (**d**) distributions.

For a more detailed analysis of the element distributions, the maps for the SEM images with higher resolution were analyzed (Figure 6). The texture of the sample in the SEM image (Figure 6a) is attributed to both its porous structure and features of its pretreatment prior to the microscopic studies. Thus, the part in white in the center of the image can be attributed to the place where the sample was broken before the microscopic studies and corresponds to the "bulk" of the sample. The part in grey can be attributed to the real surface of the sample. Figure 6b shows a uniform Si distribution on the surface. However, if we combine the maps for the O, Mg, Al, and Si distributions (Figure 6c), we can see that Mg and Al are distributed predominantly in the bulk of the sample, while the surface is enriched with Si.

Thus, the combination of oxalic acid and NaOH treatment of cordierite leads to the Mg and Al leaching from the surface with the formation of a Si-rich surface layer with a thickness of about a few micrometers. This silica layer predominantly participates in the stabilization of the supported active components. It can be seen from the map of Mn, Ni, and Cu distributions (Figure 6d) that their concentrations are lower in the bulk of the sample and higher on the surface.

### 3.3. Features of the Phase Composition of the Samples

The phase composition of the catalysts was studied by powder XRD. All reflections in the X-ray diffraction patterns for initial cordierite and pretreated supports Cor-$2H_2C_2O_4$ and Cor-optimized (Figure 7a) correspond to the cordierite crystal structure (Card No. 12-0303), which indicates that etching with oxalic acid and/or NaOH does not lead to significant changes in the crystal support structure. The low-intensity halo at 17–28 2θ may be attributed to the presence of impurities of the amorphous phase in the sample. The growth of intensity of this halo is observed for pretreated supports (see insert in Figure 7a), which may indicate the formation of an amorphous phase on the cordierite surface. According to the SEM results, this phase is enriched with $SiO_2$, and this phase probably provides a significant rise in the surface area for the pretreated supports.

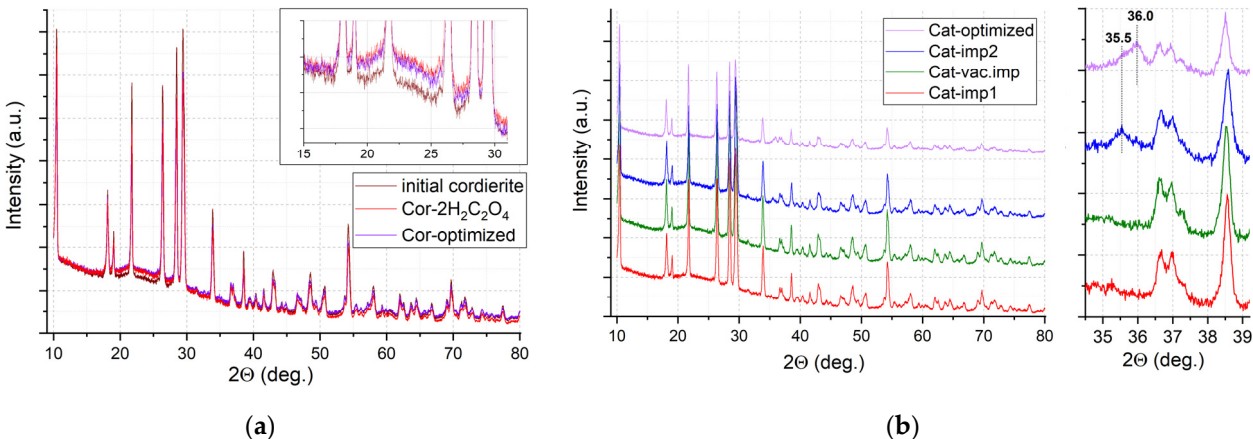

**Figure 7.** XRD patterns for supports (**a**) and catalysts (**b**).

Similar XRD patterns are observed for catalysts: for Cat-imp1 and Cat-vac.imp, all XRD reflections in the region of 2θ angles are referred to as the cordierite crystal structure (Figure 7b). The absence of reflections from the supported oxides may be attributed to their low loading and/or highly dispersed, weakly crystallized state. However, after the second impregnation (sample Cat-imp2) and for Cat-optimized, a weak reflection appears at 2θ = 35.5 that can be attributed to $Mn_2O_3$ (Card No. 33-0900), or copper or nickel manganites (Card No. 32-0345), and a weak reflection appears at 2θ = 36.0 deg. for Cat-optimized, which is probably associated with the formation of the tetragonal phase of $Mn_3O_4$ (Card No. 24-0734).

### 3.4. Temperature-Programmed Reduction Results

The features of the sample reduction were studied by the TPR-$H_2$ method (Figure 8). For the support, hydrogen consumption is not observed in the entire temperature range, which confirms the high thermal stability in the reducing medium of the cordierite block. All catalysts are characterized by the presence of a hydrogen consumption peak from 150 °C to 320 °C, and a small shoulder in the temperature range of 320–550 °C. The TPR profile is not similar due to the reduction in manganese oxides ($MnO_2$, $Mn_3O_4$, and $Mn_2O_3$), which is usually characterized by two separated intensive TPR peaks at higher temperatures. The peak at 245–262 °C can be attributed to the reduction in dispersed $Cu^{2+}$ species accompanied by the reduction in Mn(III, IV) and Ni(II) [27].

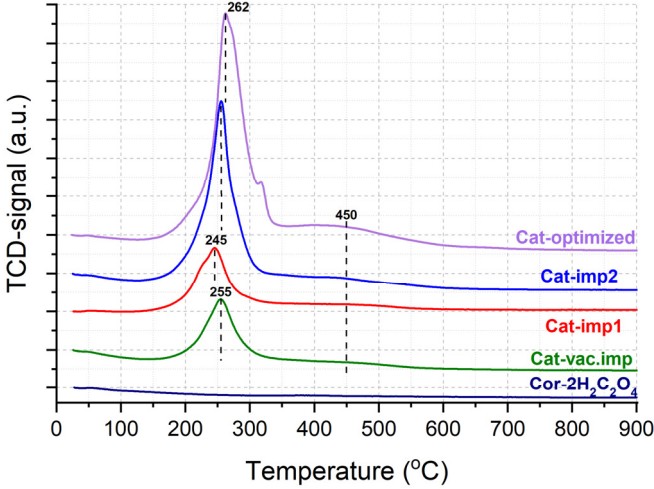

**Figure 8.** TPR-$H_2$ profiles for pretreated support and synthesized catalysts.

A broad peak at 450 °C can be attributed to the reduction in $Cu^{2+}$ and $Ni^{2+}$ species strongly bound to the support surface with a possible formation of copper and nickel silicates [28]. With an increase in the content of deposited oxides from 1.8 wt.% for Cat-vac.imp up to 7 wt.% for Cat-optimized, the intensity of hydrogen consumption increases (Table 4). Thus, it can be concluded from TPR-$H_2$ that the active components are mainly stabilized on the surface as mixed oxides or as interacting particles. A part of the components is strongly bonded with the cordierite surface, which has a high impact on the stabilization of supported components and high mechanical stability towards vibration (Table 3). The addition of Ni and Cu leads to the increased reducibility of the Mn oxides, which is favorable for ozone decomposition at low temperatures according to Equations (1) and (2).

**Table 4.** Results of TPR-$H_2$.

| Samples | $H_2$ Consumption (mmol/g) | | | |
|---|---|---|---|---|
| | at 150–350 °C | at 350–600 °C | $\Sigma$ | Reduction Degree of Oxides, % |
| Cat-vac.imp | 0.185 | 0.012 | 0.197 | 70.8 |
| Cat-imp1 | 0.182 | 0.014 | 0.196 | 91.6 |
| Cat-imp2 | 0.386 | 0.029 | 0.415 | 89.8 |
| Cat-optimized | 0.538 | 0.110 | 0.648 | 76.3 |

*3.5. Activity of Catalysts in Ozone Decomposition*

The catalytic properties of the synthesized catalysts were studied in the reaction of ozone decomposition (Figure 9). The catalysts were tested at room temperature at flow rates of 20 or 50 L/min and initial ozone concentrations of 1 or 2 ppm. The initial concentration of ozone was recorded at the reactor inlet, while the final concentration was determined at the reaction outlet. The ozone conversion for the support pretreated with oxalic acid is near 0% at an airflow rate of 20 L/min. Thus, the catalytic activity can be attributed to the ozone decomposition on the active component and not on the reactor and cordierite walls.

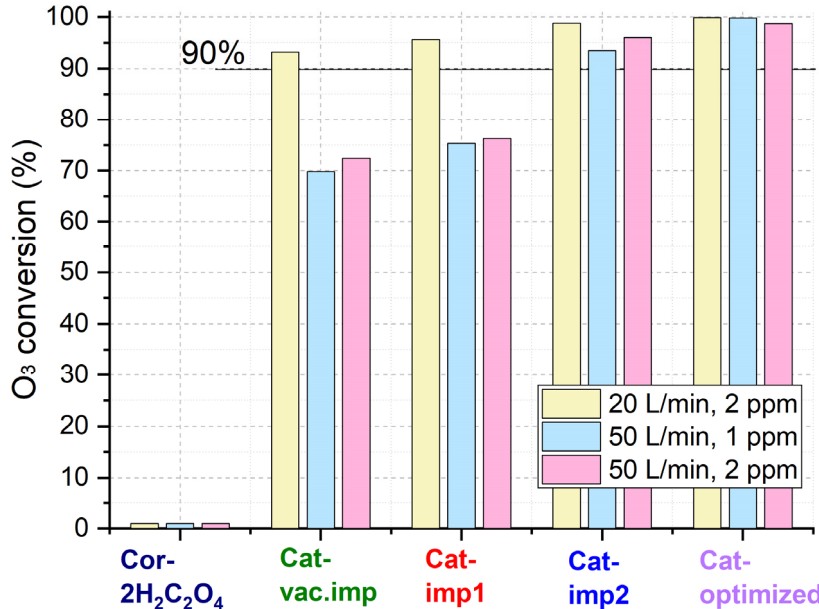

**Figure 9.** $O_3$ conversion over synthesized catalysts.

At a flow rate of 20 L/min and an initial ozone concentration of 2 ppm, with an increase in the content of active components, the conversion increases from 93% for Cat-vac.imp to 100% for Cat-optimized (Table 5). As the airflow rate increases up to 50 L/min, the ozone conversion over the catalysts decreases. Despite this, the catalysts pre-

pared under optimized conditions reach high conversions (>98%) at ozone concentrations of 1 and 2 ppm.

**Table 5.** Catalytic properties of the samples at room temperature.

| Flow Rate (L/min) | Sample | $O_3$ Concentration (ppm) | Conversion (%) |
|---|---|---|---|
| 20 | Cor-2H$_2$C$_2$O$_4$ | 2.0 | 0.0 |
| | Cat-vac.imp | 2.0 | 93.2 |
| | Cat-imp1 | 2.1 | 95.6 |
| | Cat-imp2 | 2.1 | 98.8 |
| | Cat-optimized | 2.1 | 99.9 |
| 50 | Cat-vac.imp | 0.9 | 69.8 |
| | | 2.0 | 72.4 |
| | Cat-imp1 | 1.0 | 75.3 |
| | | 2.1 | 96.0 |
| | Cat-imp2 | 0.9 | 93.5 |
| | | 2.1 | 96.0 |
| | Cat-optimized | 0.8 | 99.8 |
| | | 2.0 | 98.7 |

The sample Cat-optimized was studied at a higher airflow rate of 150 L/min when varying the temperature from 25 °C to 120 °C (Figure 10). It was shown that with an increase in the temperature from 20 °C to 120 °C, the ozone conversion over the Cat-optimized catalyst increases from 92% to almost 96%. The ozone conversion at such a high gas velocity is determined by both the activity of catalysts and the features of the gas flow inside the catalyst channels. The combination of the porous structure of catalysts and the highly dispersed state on the supported active transition metal oxides allows for achieving conversion above 90% at temperatures from 20 to 120 °C.

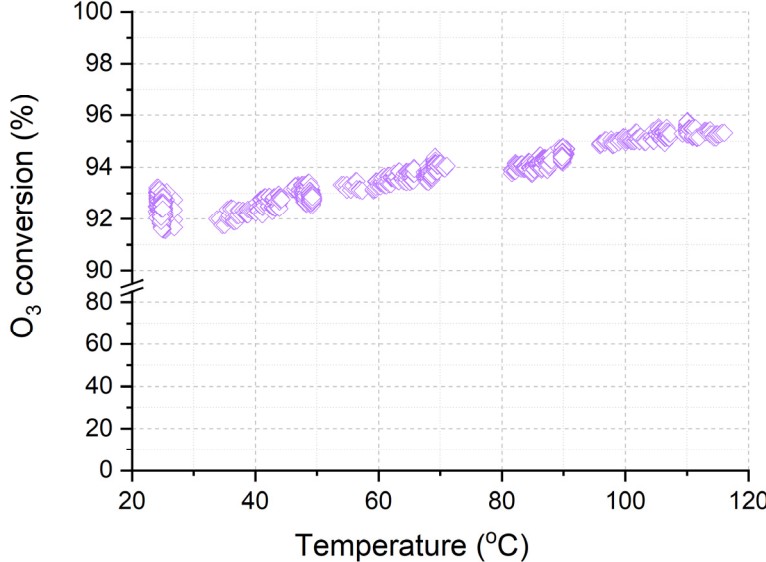

**Figure 10.** Temperature dependence of $O_3$ conversion over Cor-optimized catalyst.

Thus, it was shown that a combination of cordierite ceramic block and supported Mn-Cu-Ni oxide composition allowed the acquisition of a catalyst with high mechanical stability, including resistance towards vibration, and high catalytic activity in ozone decomposition even at high gas velocities. The porous structure of the catalysts played an important role in both the catalytic activity and stabilization of supported active components. It was found that the cordierite treatment with acid solutions led to the formation of micropores, while

the etching with alkali resulted in the formation of mesopores. Both micro- and mesopores were formed during the combined cordierite treatment with $H_2C_2O_4$ and NaOH solutions. The specific surface area of cordierite can be increased up to ~100 $m^2$/g by treatment in acids, while the subsequent increase in $S_{BET}$ led to a decrease in the mechanical strength.

The cordierite-supported catalysts were synthesized by the impregnation methods with the loading of a mixture of $MnO_2$, CuO, and NiO oxides from 1.8 to 7.0 wt.%. It was shown that the synthesized catalysts featured a specific surface area of 25–63 $m^2$/g, and the active components were uniformly distributed in the pores and on the support surface in a highly dispersed, weakly crystallized state. The Si-rich surface of cordierite ceramics after the acid treatment led to stabilization of the supported active oxides.

All catalysts show high activity in ozone decomposition at room temperature at a high gas flow rate of 20–50 L/min and initial ozone concentrations of 1–2 ppm. The high activity of the catalysts is mainly attributed to both the relatively high surface area of the catalysts and the highly dispersed state of the supported oxides. The combination of Mn, Cu, and Ni oxides leads to an enhanced reaction ability of the surface in comparison with the individual $MnO_x$. The TPR data show a significant shift in the reduction temperature of $MnO_x$ in the presence of Cu and Ni. In the catalytic reaction, it should lead to an easy reduction and re-oxidation of these Mn species by ozone, even at room temperature (Equations (1) and (2)), because of the rather high reaction ability of the $O_3$ molecule.

An increase in the $O_3$ conversion is observed with an increase in the content of the active components and a respective increase in the number of active species (as shown by TPR). The highest activity was observed for the Cat-optimized catalysts, which was characterized by a mixed micro–mesoporous structure that allowed increasing the loading of active components up to 7.0 wt.% while keeping both high mechanical strength and stability to vibrations. It was shown that the double impregnation led to a decreased stability to vibrations. Thus, optimization of the support pretreatment allows the increase in the pore volume and moisture capacity that is necessary to stabilize a higher amount of supported active components. The relatively high loading of the active components (7.0 wt.%) is required for both higher activity and stability for long-term exploitation of the catalyst.

The synthesized catalyst is promising for ozone decomposition and can be used in air purification systems in transport, working zones, and other public places because it is noble-metal-free and is characterized by high activity even at very high gas velocities, high mechanical stability, and low hydrodynamic resistance due to the honeycomb structure.

## 4. Conclusions

The Mn-Cu-Ni oxide catalysts supported on cordierite ceramics were suggested as highly active materials for ozone decomposition that did not contain expensive noble metals. It was shown that a combination of cordierite ceramics, and their treatment with acids and alkali, provided the preparation of catalysts with developed micro–meso–macroporous structures that played an important role in both the stabilization of supported active components and high catalytic activity. The ozone conversion above 98% was achieved at room temperature over Cat-optimized block catalysts with a flow rate of 50 L/min and ozone concentration of 2 ppm. The conversion above 92% was observed for this catalyst at a flow rate of 150 L/min.

**Author Contributions:** M.C.: methodology, investigation, writing—original draft preparation, and visualization; M.G.: investigation and writing—original draft preparation; A.K.: conceptualization, methodology, and resources; G.M.: conceptualization, methodology, resources, writing—review and editing, supervision, and funding acquisition. All authors have read and agreed to the published version of the manuscript.

**Funding:** This research was funded by State assignment of the Ministry of Education and Science of the Russian Federation (project number FSWM-2020-0037).

**Data Availability Statement:** The data presented in this study are available on request from the corresponding author.

**Acknowledgments:** The authors thank Tomsk regional collective use center (Tomsk State University) for SEM studies.

**Conflicts of Interest:** The authors declare no conflict of interest.

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
