# Peer review of "Cordierite-Supported Transition-Metal-Oxide-Based Catalysts for Ozone Decomposition"

_crystals, doi:10.3390/cryst13121674_

Round 1
Reviewer 1 Report
Comments and Suggestions for Authors
Maria et al treated the cordierite ceramic with oxalic acid and NaOH, and prepared the cordierite-based supported Mn-Cu-Ni catalysts by the impregnation method and used them for ozone decomposition. The manuscript was written well, however, some questions are needed more explanation. I recommend a minor revision before acceptance. The comments are listed as follows.
1. Fig. 2, it is found that the NaOH treated cordierite shows a significantly different N2 adsorption-desorption isotherm compared to the other samples, please give more explanation.
2. For the TPR results, is it possible to calculate the reduction degree of each sample? Will the reduction of the catalyst affect the O3 conversion?
3. Which one is the active center for the conversion of O3? Please give the reason for the selection of Cu, Mn and Ni in the introduction.
4. The loading of the metal species is low in the catalyst, is it accurate enough to measure the XRD patterns?
5. It is better to show more about the relationship between the characterization results and the O3 conversion.
Author Response
Dear Reviewer, thank you for your interest to our study and positive recommendations. We improved the manuscript according to your comments and remarks. All changes in the manuscript are highlighted in blue.
Maria et al treated the cordierite ceramic with oxalic acid and NaOH, and prepared the cordierite-based supported Mn-Cu-Ni catalysts by the impregnation method and used them for ozone decomposition. The manuscript was written well, however, some questions are needed more explanation. I recommend a minor revision before acceptance. The comments are listed as follows.
- Fig. 2, it is found that the NaOH treated cordierite shows a significantly different N2 adsorption-desorption isotherm compared to the other samples, please give more explanation.
Response: Thank you for your comment. According to our results, the mechanism of acid and alkali activation is different. The Al and Mg leaching under acid conditions leads to the formation of micropores, while the Si leaching from the cordierite in alkali conditions leads to the formation of meso- and macropores. The alkali activation occurs with both transport of soluble silica (Na2SiO3(aq)) through the solution and along the surface, that is accompanied by the re-deposition of silica on the cordierite surface. The corresponding remarks were added into the manuscript (page 7).
- For the TPR results, is it possible to calculate the reduction degree of each sample? Will the reduction of the catalyst affect the O3 conversion?
Response: Thank you for your comment. The reduction degree of the deposited oxides was calculated, the results are shown in the table below, and the corresponding column was added into Table 4.
|
Sample |
n(Me) calculated according to TPR-H2, mmol/g |
n(Me) theoretical, mmol/g |
reduction degree, % |
|
Cat-imp1 |
0.196 |
0.277 |
70.8 |
|
Cat-vac.imp |
0.197 |
0.215 |
91.6 |
|
Cat-imp2 |
0.415 |
0.462 |
89.8 |
|
Cat-optimized |
0.648 |
0.849 |
76.3 |
These values are not close to 100 % because of a relatively high experimental error of TPR method. The real chemical state of oxides (including possible formation of Mn2+, Mn4+, Cu+, etc.) is also not clear.
The TPR results indicate the reactivity of the oxides. The reduction ability of the catalysts plays a key role in the mechanism of ozone decomposition since the reaction proceeds due to the transfer of electrons between the catalyst surface and О3 molecules. According to the literature data (reactions 1 and 2), the supported Men+ should have a high ability to oxidation into Men+1 and recombination to their original state Men+. This will determine the activity of the catalyst in ozone decomposition
(1)
(2)
The starting temperature of the catalyst reduction is more important than the reduction degree of oxides since the low-temperature active species can participate in the O3 decomposition. Thus, according to the TPR results, we observe a significant shift of the MnOx reduction peak towards low temperatures in the presence of Cu and Ni. Thus, increased reducibility of MnOx affects the activity in O3 decomposition.
Corresponding comments were added in the text (pages 2, 10 and 11).
- Which one is the active center for the conversion of O3? Please give the reason for the selection of Cu, Mn and Ni in the introduction.
Response: Thank you for your comment. According to the literature, the ozone conversion proceeds over the Mn-related species (equations (1) and (2)). Selection of Mn is linked to its highest activity in the ozone decomposition among the transition metal oxide. Only noble metals are active. The choice of Cu, Ni is linked to the increased activity of MnOx with their presence according to the literature. Corresponding comments were added in the text (page 2).
- The loading of the metal species is low in the catalyst, is it accurate enough to measure the XRD patterns?
Response: Thank you for your comment. We agree that the loading of the supported oxide is relatively low, and these oxide phase cannot be detected reliably by XRD. On the other hand, the ceramic-based catalysts have an “egg-shell” structure, and the active components are distributed predominantly on the external surface that was shown by SEM and mapping. Thus, the real surface concentration of the oxide is relatively high and is enough to catalyze the ozone decomposition reaction.
- It is better to show more about the relationship between the characterization results and the O3 conversion.
Response: Thank you for your comment. The relationship between the catalyst structures and activities was added into the discussion part (page 13).
Reviewer 2 Report
Comments and Suggestions for Authors
The report describes investigations on the pretreatment of cordierite ceramic honeycomb supports with several acids and bases, the structural/morphological consequences in particular in view of surface area and porosity, and the additional impregnation with a mixed Mn-Cu-Ni oxide active phase to form very active catalysts for the decomposition of ozone. These tests clearly demonstrate together with a serious number of characterization techniques that a combined treatment of cordierite with oxalic acid and NaOH leads to increased porous structure at the outermost surface of the shaped cordierite skeleton and additionally to stabilization of dispersed Mn-Cu-Ni oxides and finally to highly active catalysts for ozone decomposition even at room temperature in particular with high space velocities.
The results are very interesting and new, important on the way to practical (industrial) application and the manuscript is well structured. The experiments are well-chosen, designed, and performed seriously. The manuscript has a clear focus. There is therefore no doubt for me that this manuscript should be published, and I have very few minor comments.
1) The authors should rethink the use of the term “activation” for the pretreatment of cordierite. In catalysis, this term is reserved in particular for the pretreatment immediately before catalysis for the transformation of the “pre-catalyst” into its active form. Targeted pretreatment or leaching would possibly be less misleading.
2) The introduction (last two paragraphs) does not really lead the reader to the very interesting, important, and crucial subject of the report. It is not (mainly) the development of a new mixed oxide for ozone decomposition or the characterization and understanding of the activity. The real power of this contribution is the detailed investigation of the cordierite pretreatment the importance of which may not be clear enough to all readers and should be described more clearly.
3) Chemical reaction equations should be given with an arrow (or equilibrium arrow) only and never by an equal sign. The dissolution of solid oxides (etching/leaching, loss into solution) can possibly be better visualized by ionic reaction products rather than with pure sum equations (3-6).
4) It is not clear to me, why this manuscript is proposed to be published in Crystals instead of Catalysts.
Author Response
Dear Reviewer, thank you for your interest to our study and positive recommendations. We improved the manuscript according to your comments and remarks. All changes in the manuscript are highlighted in blue.
The report describes investigations on the pretreatment of cordierite ceramic honeycomb supports with several acids and bases, the structural/morphological consequences in particular in view of surface area and porosity, and the additional impregnation with a mixed Mn-Cu-Ni oxide active phase to form very active catalysts for the decomposition of ozone. These tests clearly demonstrate together with a serious number of characterization techniques that a combined treatment of cordierite with oxalic acid and NaOH leads to increased porous structure at the outermost surface of the shaped cordierite skeleton and additionally to stabilization of dispersed Mn-Cu-Ni oxides and finally to highly active catalysts for ozone decomposition even at room temperature in particular with high space velocities.
The results are very interesting and new, important on the way to practical (industrial) application and the manuscript is well structured. The experiments are well-chosen, designed, and performed seriously. The manuscript has a clear focus. There is therefore no doubt for me that this manuscript should be published, and I have very few minor comments.
1) The authors should rethink the use of the term “activation” for the pretreatment of cordierite. In catalysis, this term is reserved in particular for the pretreatment immediately before catalysis for the transformation of the “pre-catalyst” into its active form. Targeted pretreatment or leaching would possibly be less misleading.
Response: Thank you for your comment. We agree that term the “activation” is usually used for the catalysts. We have replaced the term "activation" with “pre-treatment” along the text. However, the cordierite treatment with alkali and acid leads to its activation because of the increase in the concentration of surface functional groups (monitored by FTIR, results are not shown in the present manuscript). Both increased surface area and functional groups leads to the increased reaction ability of cordierite towards metal oxide precursor. It is rather important for stabilization of the supported component. Non-activated cordierite does not allow stabilizing the active components, and the catalysts are characterized by the low stability to vibration.
2) The introduction (last two paragraphs) does not really lead the reader to the very interesting, important, and crucial subject of the report. It is not (mainly) the development of a new mixed oxide for ozone decomposition or the characterization and understanding of the activity. The real power of this contribution is the detailed investigation of the cordierite pretreatment the importance of which may not be clear enough to all readers and should be described more clearly.
Response: Thank you for your comment. The role of cordierite activation by supporting of the secondary support or by etching was added into the introduction section (page 2).
3) Chemical reaction equations should be given with an arrow (or equilibrium arrow) only and never by an equal sign. The dissolution of solid oxides (etching/leaching, loss into solution) can possibly be better visualized by ionic reaction products rather than with pure sum equations (3).
Response: Thank you for your comment. The text was revised accordingly. Soluble compound was shown as (aq).
4) It is not clear to me, why this manuscript is proposed to be published in Crystals instead of Catalysts.
Response: Thank you for your comment. The selection of the journal is connected with the invitation from the Quest Editor, and the ceramic materials are in the scope of the Crystals journal.
Reviewer 3 Report
Comments and Suggestions for Authors
The authors synthesized the cordierite-supported Mn-Cu-Ni oxide catalysts, characterized their partial physicochemical properties, and evaluated their catalytic activities for ozone decomposition. This work contains some new results and could be considered for publication. However, the authors should revise their manuscript before acceptance for publication according to the following comments:
1. How does a combined treatment of cordierite with oxalic acid and NaOH lead to the developed porous structure?
2. The typical activity data as well as the Mn/Cu/Ni molar ratio and Mn-Cu-Ni oxide loading of the typical catalyst should be given in the abstract.
3. What are the active sites for the addressed reaction?
4. What are the roles of the oxides of Mn, Cu, and Ni in catalyzing the addressed reaction?
5. What are the catalytic mechanisms?
6. A comparison on catalytic activity of the as-obtained typical sample should be made with those of the related samples reported in the literature.
Comments on the Quality of English LanguageThere are some inappropriate English words or expressions in the manuscript. The authors should carefully polish the English of the whole manuscript.
Author Response
Dear Reviewer, thank you for your interest to our study and positive recommendations. We improved the manuscript according to your comments and remarks. All changes in the manuscript are highlighted in blue.
The authors synthesized the cordierite-supported Mn-Cu-Ni oxide catalysts, characterized their partial physicochemical properties, and evaluated their catalytic activities for ozone decomposition. This work contains some new results and could be considered for publication. However, the authors should revise their manuscript before acceptance for publication according to the following comments:
- How does a combined treatment of cordierite with oxalic acid and NaOH lead to the developed porous structure?
Response: Thank you for your comment. As it was shown, the initial cordierite features only macropores (SEM results) and rather low surface area (0.04 m2/g). The treatment with acid leads to leaching of Al and Mg with the formation of channels (micropores) that leads to a significant growth of the surface area. The treatment with NaOH leads to the Si leaching and the formation of meso- and macropores. Thus, we combine the acid and alkali treatments to develop porous structure (including micro-, meso- and macropores) of cordierite. Porous structure plays an important role in both stabilization of the supported oxide and in the catalytic activity.
- The typical activity data as well as the Mn/Cu/Ni molar ratio and Mn-Cu-Ni oxide loading of the typical catalyst should be given in the abstract.
Response: Thank you for your comment. The information was added.
- What are the active sites for the addressed reaction?
Response: Thank you for your comment. According to the literature, the reaction of ozone decomposition occurs over the Mnn+ species. The corresponding comments were added into the introduction section (page 2).
- What are the roles of the oxides of Mn, Cu, and Ni in catalyzing the addressed reaction?
Response: Thank you for your comment. Mn is the main active component, Cu and Ni addition leads to the increased reaction ability of MnOx (that was shown by TPR) and increased activity in ozone decomposition. The information was added into the introduction section (page 2) and to discussion (page 13).
- What are the catalytic mechanisms?
Response: Thank you for your comment. The following information was added into the introduction section (page 2):
The high activity of manganese oxides in the O3 decomposition is due to the redox ability of Mnn+ to be converted into Mnn+1 [6]:
(1)
(2)
- A comparison on catalytic activity of the as-obtained typical sample should be made with those of the related samples reported in the literature.
Response: Thank you for your comment. We agree that the comparison with other catalysts is necessary, but the conditions of catalytic experiments are rather different, thus, making such a comparison incorrect. Thus, in a number of publications, the powder of catalysts, rather high concentrations of ozone (from 20 to 10,000 ppm), and relatively low gas flows are used. In the present study we use near-real conditions including rather high gas velocity, blocking the catalysts with low gas dynamic resistance, and low concentrations of ozone (1-2 ppm) which are more realistic for the real systems. We can find the same conditions in some patents from manufacturers of industrial Pd-containing catalysts. The advantage of our catalysts is the absence of Pd as an active component.
There are some inappropriate English words or expressions in the manuscript. The authors should carefully polish the English of the whole manuscript.
Response: Thank you for your comment. The English was polished by a native speaker.